# Maintain Efficacy and Spare Toxicity: Traditional and New Radiation-Based Conditioning Regimens in Hematopoietic Stem Cell Transplantation

**DOI:** 10.3390/cancers16050865

**Published:** 2024-02-21

**Authors:** Irene Dogliotti, Mario Levis, Aurora Martin, Sara Bartoncini, Francesco Felicetti, Chiara Cavallin, Enrico Maffini, Marco Cerrano, Benedetto Bruno, Umberto Ricardi, Luisa Giaccone

**Affiliations:** 1Allogeneic Transplant and Cellular Therapy Unit, Division of Hematology, Department of Oncology, University Hospital A.O.U. “Città della Salute e della Scienza di Torino”, University of Torino, 10126 Torino, Italy; idogliotti@cittadellasalute.to.it (I.D.); aurora.martin@edu.unito.it (A.M.); benedetto.bruno@unito.it (B.B.); 2Department of Molecular Biotechnology and Health Sciences, University of Turin, 10126 Torino, Italy; 3Department of Oncology, University of Turin, 10126 Torino, Italy; mario.levis@unito.it (M.L.); sbartoncini@cittadellasalute.it (S.B.); chia.cavallin@gmail.com (C.C.); umberto.ricardi@unito.it (U.R.); 4Division of Oncological Endocrinology, Department of Oncology, University Hospital A.O.U. “Città della Salute e della Scienza di Torino”, 10126 Torino, Italy; ffelicetti@cittadellasalute.to.it; 5Hematology Institute “Seràgnoli”, IRCCS Azienda Ospedaliero-Universitaria di Bologna, 40138 Bologna, Italy; enrico.maffini@aosp.bo.it; 6Division of Hematology, University Hospital A.O.U. “Città della Salute e della Scienza di Torino”, 10126 Torino, Italy; mcerrano@cittadellasalute.to.it

**Keywords:** hematopoietic stem cell transplantation (HSCT), leukemia, conditioning regimen, total body irradiation (TBI), total marrow irradiation (TMI), total marrow and lymphoid irradiation (TMLI), radiation toxicities

## Abstract

**Simple Summary:**

We discussed the potential long-term toxicities associated with total body irradiation, its current indications, and the technical advances in radiotherapy that have resulted in the development of total marrow irradiation and total marrow and lymphoid irradiation, which might change the role of radiation-based conditioning regimens in allogeneic hematopoietic stem cell transplantation.

**Abstract:**

Novelty in total body irradiation (TBI) as part of pre-transplant conditioning regimens lacked until recently, despite the developments in the field of allogeneic stem cell transplants. Long-term toxicities have been one of the major concerns associated with TBI in this setting, although the impact of TBI is not so easy to discriminate from that of chemotherapy, especially in the adult population. More recently, lower-intensity TBI and different approaches to irradiation (namely, total marrow irradiation, TMI, and total marrow and lymphoid irradiation, TMLI) were implemented to keep the benefits of irradiation and limit potential harm. TMI/TMLI is an alternative to TBI that delivers more selective irradiation, with healthy tissues being better spared and the control of the radiation dose delivery. In this review, we discussed the potential radiation-associated long-term toxicities and their management, summarized the evidence regarding the current indications of traditional TBI, and focused on the technological advances in radiotherapy that have resulted in the development of TMLI. Finally, considering the most recent published trials, we postulate how the role of radiotherapy in the setting of allografting might change in the future.

## 1. Introduction

Total body irradiation (TBI) has historically been an integral component of conditioning to allogeneic hematopoietic stem cell transplantation (HSCT) [1].

Over time, many developments have occurred and changed the landscape of allogeneic HSCT, impacting the role of TBI as well. New drugs have been introduced as part of the conditioning regimens, and indications for HSCT have changed with the development of different biological treatment approaches to hematological diseases; at the same time, the increasing use of reduced-intensity conditioning regimens (RIC) has made HSCT more accessible to aged patients and those with comorbidities.

In this context, novel approaches to radiation-based conditioning regimens are being implemented in clinical practice to maintain the high antineoplastic efficacy of classical TBI while sparing potential long-term toxicities. In recent years, the development of highly conformal delivery techniques, such as intensity-modulated radiotherapy (IMRT) and volumetric modulated arc therapy (VMAT), have enabled the development of alternative techniques to TBI, such as total marrow irradiation (TMI) and total marrow and lymphoid irradiation (TMLI). Many centers are currently using protocols with these targeted forms of TBI, although data about their long-term toxicity are still very scarce [2].

Here, we reviewed the main findings regarding TBI-associated side effects, the impact of TBI in disease eradication, and innovative approaches to TBI.

## 2. Total Body Irradiation

### 2.1. Myeloablative Total Body Irradiation

Many studies have been published on the role of myeloablative TBI in the conditioning regimen for acute leukemia patients; however, many controversies remain, given the extreme heterogeneity of the data analyzed [3,4,5]. Furthermore, current TBI practices differ substantially across different institutions regarding the total doses delivered, the fractioning schedules, dose rates, and organ shielding.

The main challenge when planning TBI is trying to deliver radiation in the most uniform way possible to an irregularly shaped target for patient-dependent factors (such as posture or habitus and individual density) that strongly influence the dose distribution across the body. The introduction of three-dimensional (3D) CT simulation has improved the optimization and the dosimetric accuracy of TBI [6]. During the treatment simulation phase, shielding equipment may be designed to reduce the dose given to at-risk organs, such as the lungs when a myeloablative dose (≥12 Gy) is prescribed.

TBI toxicities include the risk of sinusoidal obstructive syndrome, gastro-intestinal toxicity, and skin rash; actually, the most frequent dose-limiting toxicity is radiation-induced interstitial pneumonitis, whose incidence has variously been reported in the literature, with rates ranging from 10 to 85% [7]. In vivo dosimetry is frequently performed during treatment to measure the dose at several points on the patient’s body, with the goal of limiting dose inhomogeneity to ±5% [8].

Nowadays, different modalities for TBI delivery exist, varying across centers and countries in terms of the prescribed dose, patient positioning and immobilization, organ shielding and RT techniques used [9].

Several tools may be used in order to offer comfort and support to the patients and to guarantee, at the same time, a certain degree of immobilization and treatment reproducibility (i.e., TBI stands, treatment couches, treatment tables).

Adding TBI to the conditioning regimen for bone marrow transplant allows one to boost “sanctuary” organs that could be at a higher risk of relapse when treated with chemotherapy alone. In particular, a testicular boost, delivered with electrons at a single dose of 4 Gy, is strongly recommended in all male with acute lymphoblastic leukemia (ALL) undergoing TBI conditioning for HCT [8,10].

Despite its wide experience in the field of myeloid neoplasms, the current role of myeloablative TBI in clinical practice is mainly focused on ALL patients (Table 1). Indeed, as of 2019, the EBMT (European group for Bone Marrow Transplant) and EWALL (European Working Group on Adult ALL) position statements strongly support the use of myeloablative TBI-based regimens for patients with ALL, a recommendation also endorsed by the American Society for Transplantation and Cellular Therapy [11,12].

Such recommendations are based mainly on the results of large retrospective studies that have demonstrated the clear benefits of radiation-based schemes over chemotherapy-only conditioning regimens for ALL patients regarding their progression-free survival (PFS) [13,14,15,16,17,18,19] and overall survival (OS) [14,19]. Indeed, for pediatric ALL patients (aged 4–21 at the time of HSCT), a clear survival benefit was shown in a randomized trial of TBI + etoposide versus chemoconditioning, due to a significantly reduced relapse incidence [20].

Also, in the context of haploidentical HSCT with post-transplant cyclophosphamide (PT-Cy), the EBMT group reported significantly reduced 2-year non-relapse mortality (NRM) and improved leukemia-free survival (LFS) for TBI-based conditionings compared to chemotherapy-based regimens; however, the survival benefit was not significant [21]. Another retrospective EBMT study showed a decreased risk of NRM with fludarabine (Flu)-TBI vs. thiotepa, busulfan and fludarabine (TBF), but an increased risk of relapse without a significant effect on survival and GVHD [22]. Interestingly, this study and another single-center experience [23] suggested that the association of fludarabine might be a feasible alternative to cyclophosphamide and/or etoposide for TBI-based myeloablative conditioning (MAC), which is also being explored outside of the haplo-HSCT setting [24].

As molecular and flow cytometry tools for the identification of minimal residual disease (MRD) in ALL were developed, the decision to continue HSCT, its timing, and the optimal conditioning intensity were questioned. Indeed, TBI regimens seem to be able to overcome the unfavorable prognostic factor given by positive pre-transplant MRD assessment: in a large retrospective study comprising over 2700 ALL patients, radiation-based conditioning was associated with longer OS, disease-free survival (DFS), and a lower relapse incidence, irrespective of the pre-HSCT MRD status [25]. Likewise, the survival advantage seems to be effective also among ALL patients with primary refractory disease or with a large tumor burden at the time of HSCT [26].

Nevertheless, there are still some controversies regarding the survival advantage offered by TBI-based regimens. Of note, a randomized study on 550 ALL patients transplanted in complete remission (CR1) and receiving Bu-Cy or TBI-Cy indicated the noninferiority of Bu-Cy, with a comparable 2-year relapse rate and NRM. Moreover, there were no differences in the regimen-related toxicity, GVHD, or late effects between the two groups [27].

**Table 1 cancers-16-00865-t001:** Myeloablative TBI in clinical practice.

	Study Design	Conditioning Regimen	Number of Patients	OS	DFS	RI	NRM	aGVHD	cGVHD
Bunin et al. [13]	Randomized prospective trial	TBI or Bu + Cy-Etoposide	22 vs. 21	67 vs. 47	58 vs. 29	32 vs. 43	9 vs. 24	25	9
Peter et al. [20]	Randomized prospective trial	Etoposide-TBI (12 Gy) vs. Flu-thiotepa-Bu/Treo	212 vs. 201	ALL	91 vs. 75	86 vs. 58	12 vs. 33	2 vs. 9	37 vs. 29
Jamy et al. [24]	Prospective trial	Flu-TBI (12 Gy)	19	68	63	7	31	26	21
Zhang et al. [27]	Prospective trial	Cy-TBI (9 Gy) vs. Bu-Cy	273 vs. 272	79 vs. 76	70 vs. 69	18 vs. 20	11 vs. 11	28 vs. 31	29 vs. 31
Granados et al. [15]	Retrospective multicenter cohort study	TBI-based vs. Bu-based regimens	114 vs. 42	NA	43 vs. 22	47 vs. 71	17 vs. 22	17 vs. 12	4 vs. 0
Kiehl et al. [16]	Retrospective multicenter cohort study	TBI (10–13.5 Gy)-based vs. Bu-based regimens	221	34	29	29	45	30	NA
Marks et al. [17]	Retrospective multicenter cohort study	Cy-TBI (<13 Gy) vs. Cy-TBI (>13 Gy) vs. etoposide-TBI (<13 Gy) vs. etoposide-TBI (>13 Gy)	217 vs. 81 vs. 53 vs. 151	74 vs. 74 vs. 71 vs. 80	68 vs. 69 vs. 67 vs. 79	23 vs. 16 vs. 9 vs. 12	9 vs. 13 vs. 23 vs. 9	29 vs. 24 vs. 30 vs. 25	23 vs. 23 vs. 19 vs. 34
Cahu et al. [19]	Retrospective multicenter cohort study	TBI-based vs. Bu-based regimens	523 vs. 78	47 vs. 28	44 vs. 25	33	25	40 vs. 27	44 vs. 30
Mitsuhashi et al. [3]	Retrospective multicenter cohort study	Cy-TBI vs. p.o. Bu-Cy vs. i.v. Bu-Cy	2028 vs. 60 vs. 42	69 vs. 56 vs. 71	62 vs. 54 vs. 47	20 vs. 21 vs. 24	18 vs. 24 vs. 20	40 vs. 37 vs. 33	37 vs. 31 vs. 40
Kebriaei et al. [4]	Retrospective multicenter cohort study	TBI-based (9–12 Gy or 13–16 Gy) vs. Bu-based regimens	819 vs. 299	53 vs. 57	48 vs. 45	28 vs. 37	25 vs. 19	12 vs. 47	55 vs. 49
Eder et al. [5]	Retrospective multicenter cohort study	Cy-TBI vs. thiotepa-based regimens	540 vs. 180	49 vs. 46	39 vs. 33	36 vs. 43	24 vs. 23	25 vs. 22	45 vs. 43
Pavlu et al. [26]	Retrospective multicenter cohort study	MAC (TBI 8–14 Gy) or RIC (TBI < 6 Gy)	86	36	28	51	20	33	32
Dholaria et al. [21]	Retrospective multicenter cohort study	TBI vs. CT-based regimens	188 vs. 239	51 vs. 57	45 vs. 37	NA	21 vs. 31	38 vs. 19	34 vs. 17
Solomon et al. [23]	Retrospective multicenter cohort study	Flu-TBI (12 Gy)	82	85	78	15	7	52	37
Swoboda et al. [22]	Retrospective multicenter cohort study	Flu-TBI vs. thiotepa-Bu-Flu	117 vs. 119	60 vs. 58	50 vs. 52	19 vs. 30	31 vs. 17	38 vs. 30	25 vs. 28

Abbreviations: OS, overall survival; DFS, disease-free survival; RI, relapse incidence; NRM, non-relapse mortality; aGVHD, acute graft-versus-host disease; cGVHD, cronic graft-versus-host disease; Cy, cyclophosphamide; TBI, total body irradiation; Bu, busulfan; Flu, fludarabine; Treo, treosulfan; MAC, myeloablative conditioning; RIC, reduced-intensity conditioning; CT, chemotherapy; ALL, acute lymphoblastic leukemia; rALL, relapsed or refractory acute lymphoblastic leukemia. NA, not applicable.

### 2.2. Reduced-Intensity Total Body Irradiation

With the aim of reducing the significant toxicities associated with full-dose myeloablative TBI, some groups have explored the possibility of a TBI-based reduced-intensity regimen [28].

Researchers from Seattle pioneered the introduction of a minimally intensive conditioning regimen, incorporating 2 Gy TBI, in order to offer allogeneic HSCT to comorbid and aged patients [29]. A phase III trial among patients with hematologic malignancies treated with 2 Gy TBI alone vs. 2 Gy TBI with fludarabine 90 mg/m^2^ determined that adding fludarabine contributed to a lower relapse risk (40% vs. 55%), resulting in superior survival (60% vs. 54% at 3 years) [30]. Thus, from 1997 to 2009, a total of 1092 consecutive patients with a variety of hematological diseases included in prospective trials received low-dose TBI, with or without fludarabine. After a median follow-up of 5 years, the 5-year survival ranged from 25% of patients with high-risk disease features to 60% for low-risk diseases, with a NRM rate of 24% and a relapse mortality rate of 34.5% overall [31]. A recent updated long-term analysis demonstrated a net prolongation of survival rates in the 2010–2017 cohort with respect to the 1997–2003 cohort, with a lower NRM incidence due to refined support measures and lower GVHD-associated morbidity and mortality rates [32].

Krakow et al. recently reported the result of a prospective phase I/II trial testing clofarabine with 2 Gy TBI in adults with AML, showing a 2-year OS and LFS of 55% and 52%, respectively. Taking into account the limitations of such a comparison, these results were superior to those of a historical high-risk cohort treated with fludarabine and 2 Gy TBI [33]. In an attempt to overcome disease recurrence, which is still a major cause of treatment failure [34,35,36], the TBI dose was escalated up to 3, 4 or 4.5 Gy in a group of 77 adult patients affected by MDS, chronic myelomonocytic leukemia or other myeloproliferative disorders. As expected, the patients receiving a higher TBI dose had a lower 5-year relapse rate (32% vs. 45%) [37].

The Seattle group tested the addition of low-dose TBI (2 Gy) to an intensive conditioning regimen including treosulfan and fludarabine in AML and MDS patients up to 70 years. The 1-year relapse incidence was 16% with the addition of TBI and 35% without (*p* = 0.05) [38]. Similar results were reported in a retrospective, single-center study conducted in 63 patients with the addition of TBI 4 Gy to a Bu-Flu regimen; the study also highlighted a lower 5-year cumulative incidence of chronic GVHD with Bu-Flu-TBI (29% vs. 52%) [39]. Interestingly, a recent single-center study compared patients who underwent allo-HSCT for primary or secondary myelofibrosis conditioned with Bu-Flu (*n* = 8) and Bu-Flu plus 2 Gy of TBI (*n* = 25), showing superior engraftment and remission rates with the addition of TBI, with a comparable OS [40].

Bornhäuser and colleagues reported the results of a phase III trial comparing a regimen with 8 Gy TBI and fludarabine with Cy-TBI 12 Gy in adult AML patients in CR1. They showed a reduction in the 12-month NRM and a reduction in early toxicities in the reduced intensity arm, while the OS, DFS and relapse incidence were comparable in the two groups [41]. Importantly, no evidence that reduced-intensity conditioning increased the risk of late relapse was observed after a median follow-up of about 10 years [42]. Finally, the EBMT ALWP retrospectively compared the outcomes of ALL-CR1 patients who underwent allo-HSCT with TBI-based conditioning at a total dose of 12-Gy vs. 8-Gy. In both the univariate and age-adjusted Cox proportional hazards analyses, the relapse, NRM, LFS, OS, GVHD-free, and relapse-free survival (GRFS) were not influenced by the TBI dose, suggesting that 12-Gy and 8-Gy result in a similar outcome [43].

## 3. TBI-Associated Late Toxicities

The exact impact of TBI on the late effects of anticancer treatments is difficult to assess, since patients who are candidates for TBI generally receive a combination of chemo-radiotherapy as a conditioning regimen and several chemotherapy agents as pre-transplant treatments [44]. While it has been demonstrated in animal models of myeloablative TBI that the dose of CD34-positive stem cells contained in the graft correlates with the rate of radioprotection [45], most clinical data derive from historical pediatric cohorts [46,47,48,49,50]. Finally, only a few studies have included new radiation therapy (RT) techniques, and these have a relatively short follow-up period [51].

Endocrine complications, particularly primary hypothyroidism, represent one of the most common late effects of HSCT [44]. Due to the direct relationship between the radiation dose and the risk of thyroid dysfunction, patients treated with TBI usually show subclinical hypothyroidism and do not require hormone replacement therapy [52,53]. In contrast, some studies have found no difference in the incidence of hypothyroidism between TBI-exposed and non-exposed patients [54,55,56]. Recently, a study performed at our center suggested that pre-transplant TSH levels seem to predict the onset of post-HSCT hypothyroidism [57]. An increased risk of abnormal glucose tolerance and reduced insulin sensitivity were reported in acute lymphoblastic leukemia (ALL) survivors who received a TBI conditioning regimen before HSCT [58]. Moreover, a reduced pancreatic volume has been found after a radiation-based conditioning regimen, suggesting that RT has a direct role in the onset of insulin resistance and diabetes after HSCT [59]. Recently, a cross-sectional cohort study demonstrated an altered production of incretin hormones in HSCT survivors previously treated with TBI, with them developing dyslipidemia and abdominal adiposity [60].

TBI has also been associated with a higher risk of dyslipidemia caused by endocrine abnormalities [61,62,63,64]. The cardiovascular (CV) risk after HSCT is the result of classical patient CV risk factors and treatment-related factors (anthracyclines exposure and/or mediastinal RT, high-dose corticosteroids and immunosuppressive drugs) [65], other than TBI-associated damage.

In patients treated with TBI, delayed interstitial lung disease has also been reported, especially in cases previously exposed to mediastinal RT or chemotherapy agents with potential lung toxicity such as bleomycin, methotrexate and carmustine [66,67,68].

Infertility is another well-recognized side effect of HSCT [69,70,71,72], since the seminiferous tubules in males and the ovaries in females are highly sensitive to the detrimental effects of alkylating agents and RT [73,74,75]. Patients exposed to TBI show a significant risk of cataracts [76] and of skeletal alteration [77], involving also dental and/or craniofacial skeletal structures, with a clear relation to the age at treatment [78]. Premature aging and chronic low-grade inflammation are thought to be crucial pathophysiological mechanisms implicated in the development of CV disease after anticancer treatments [79]. In pediatric childhood cancer survivors, TBI has been found to be a major risk factor for frailty and sarcopenia, which are typical features of premature aging phenotypes [80]. Moreover, higher levels of inflammatory cytokines and advanced glycation end-products have been reported in long-term HSCT survivors treated with TBI [81].

In addition to the risk of the recurrence of the underlying disease (that progressively decreases over time), HSCT survivors show an increased risk of malignancies when compared to the general population. The risk of a second malignant neoplasm varies depending on the cohort considered, the follow-up duration and the age of the participant at the time of HSCT, without reaching a plateau even many decades after transplant [82,83,84]. TBI can play a significant role in the development of thyroid [85] and breast cancers [86], as well as of oral cavity and skin tumors [87]. For patients receiving cranial irradiation, there is also an increased risk of brain tumors, particularly if HSCT was performed at a younger age and with higher irradiation doses, as was demonstrated in long-term survivors of childhood acute leukemia [88].

## 4. Total Marrow and Total Lymphoid Irradiation

Technological advances in modern RT have not been applied to TBI for many decades. In fact, the traditional methods of TBI planning and delivery, with large, opposed whole-body fields and no beam conformation, have not changed in the last 30 years [8]. This planning translates to a high level of dose heterogeneity, frequently exceeding 30%, which is considered unacceptable when conformal techniques are applied [89]. Despite the adoption of lung shielding, conventional TBI does not allow the lung dose to be kept below the threshold dose of 8 Gy, which is a well-recognized dose constraint used to further reduce lung toxicity (lethal pneumonitis) and to consequently improve OS [90].

As a consequence, alternative non-TBI containing regimens have been increasingly tested to replace radiation and to reduce toxicity [91]. Therefore, the role of TBI in the conditioning phase of HSCT was reconsidered, taking into account all the challenges of such a poorly conformal technique.

The technological advances in modern RT allow for the delivery of highly conformed radiation beams, even to large body volumes. In particular, Intensity-Modulated Radiotherapy (IMRT), Image-Guided Radiotherapy (IGRT) and Helical Tomotherapy (HT) provide a more selective irradiation of bone marrow, lymph nodes and circulating blood, with better control of the radiation dose delivery.

The combination of all the mentioned advancements resulted in the development of total marrow irradiation (TMI) and total marrow and lymphoid irradiation (TMLI). These represent methods able to deliver organ-sparing targeted TBI using IMRT, HT in particular, and IGRT, thus providing the more selective irradiation of bone marrow, lymph nodes and circulating blood, with better control of the radiation dose delivery. HT represents, by far, the most advanced RT platform available for the delivery of TMI and TMLI (Figure 1).

### 4.1. Planning

The TMI/TMLI treatment is delivered in supine position with immobilization devices (thermoplastic mask for head and shoulders combined with a customized vacuum pillow) [92] (Figure 2). Two separate CT scans are required to cover the entire body of the patient.

Further, a 4-dimensional CT scan (4DCT) or deep inspiration breath-holding (DIBH) CT scan can be acquired to compensate for respiratory-related organ motion.

The clinical target volume (CTV) for TMI is skeletal bone, with the addition of the major lymph node chains of the spleen and liver (and, eventually, “sanctuary sites” like testes and brain) for TMLI. Colleagues from City of Hope, who first developed the TMI technique [51], suggest omitting the mandible in the CTV to reduce the likelihood of severe oral mucositis [93]. The expansion margins from the CTV to the planning target volume (PTV) vary across institutions (between 5 and 10 mm) [94,95,96,97]. The organs at risk usually include the brain (when not included in the target), eyes, lens, optic nerves and chiasm, oral cavity, major salivary glands, thyroid, esophagus, lungs, heart, breasts, stomach, liver, spleen (when not included in the target), kidneys, small and large intestine, rectum, bladder and genitals. Table 2 shows a comparison of the median dose received by at-risk organs with TBI and with TMI/TMLI, while Figure 3 shows the contours of at-risk organs in representative CT slices.

### 4.2. Prescription Dose and Fractionation

The most common TMI/TMLI fractionation schedules use fractions of 1.5–2 Gy, administered twice a day. The total prescription dose was 12 Gy in the original reports [51], but recent studies have tested schedules of 20 Gy, with preliminary reports on the safety of this dose escalation, when applied at 2 Gy fractions [98,99,100,101]. Larger fraction doses can limit dose escalation due to increased acute toxicities [100].

### 4.3. Treatment Delivery

The delivery of TMI and TMLI requires the adoption of intensity-modulated radiation therapy (IMRT) planning solutions, which employ multiple segmented and modulated beams to accurately carve the radiation dose, even to tumors with highly irregular shapes.

TMI treatments are mostly delivered with HT [51]. HT integrates the technological advances of computed tomography (CT) IGRT and the helical delivery of IMRT in a single device.

Initial planning comparison studies demonstrated that TMI is able to keep the median doses to the healthy organs (such as the brain, lungs, heart, kidneys, small intestine, liver) approximately to 40–60% of the prescribed dose to the target, with a potential relevant reduction in the acute and chronic toxicity profile compared to standard TBI [51,102,103,104], as shown in Table 2.

A recent publication from Shinde et al. showed a reduction in chronic toxicity with TMI/TMLI compared to historical cohorts treated with conventional TBI in a group of long-term survivors treated at City of Hope (median follow-up of 5.5 years). The cumulative incidence of infection and radiation pneumonitis was 23% at 2 years after TMI/TMLI, with a mean lung dose of 8 Gy or more being the strongest predictor of pulmonary complications [105].

### 4.4. TMI/TMLI Indications and Current Role

An increasing number of centers worldwide have initiated the replacement of TBI with TMI and TMLI in the context of clinical trials, given the investigational nature of these innovative RT strategies. The future adoption of TMI/TMLI into the clinical routine is largely dependent on the results of these ongoing studies, demonstrating superior outcomes in terms of a reduced toxicity profile and/or improved tumor control and cure rate, compared to standard TBI.

The first phase I and pilot studies focused on high-risk patients with advanced disease or those who were not candidates for standard HCT regimens. Given the encouraging results of these studies, phase II trials were launched in patients with less advanced diseases. More recent and ongoing trials are testing TMI/TMLI-containing regimens for the replacement of TBI in the standard-risk population.

The current strategies actively investigated are detailed below.

*(a)* 
*Dose escalation of TMI/TMLI to improve disease control in high-risk patients who have a poor outcome with standard HSCT protocols.*


Two phase I trials from City of Hope concluded that RT dose escalation is not feasible in combination with chemotherapy regimens containing busulfan and etoposide because of toxicity [106]. The same authors in another phase I trial concluded that the conditioning regimen with TMLI 20 Gy/CY/Etoposide(VP-16) was feasible (CY dose: 100 mg/kg, VP-16 dose: 60 mg/kg), with an acceptable toxicity profile and promising control rates in a very high-risk population [98]. A phase II trial with the same regimen is currently ongoing (trial number NCT02094794), with a primary endpoint of PFS at 2 years. Larger fraction sizes were investigated in a couple of phase I studies [100].

Preliminary clinical experiences demonstrate that dose escalation is feasible, but the maximum tolerated dose is influenced by many factors, like the fractionation dose/schedule, the chemotherapy regimen used, and the timing of chemotherapy with the delivery of TMLI. To date, the highest dose escalation (20 Gy) was achieved with a standard fractionation schedule (2 Gy/BID), combined with a CY/VP-16 regimen given after radiation [98].

*(b)* 
*Integration of TMI/TMLI in reduced-intensity conditioning regimens to improve disease control without increasing the toxicity profile*


Less intensive conditioning regimens have been developed in order to offer HCT to unfit patients [107,108,109,110]. TMI/TMLI then has the potential to improve the outcome, with similar toxicities, when in combination with chemotherapy in the setting of RIC. Rosenthal et al. combined TMLI (12 Gy in 1.5 Gy/BID, days −7 to −4) with the established RIC regimen of Flu (25 mg/m^2^ for 5 days) and melphalan (140 mg/m^2^ for 1 day) in patients older than 50 years and ineligible for myeloablative regimens. All patients engrafted and no increased toxicity was detected [104]. The update of this study also showed promising long-term results, with 5-year OS and EFS rates of 42% and 41%, respectively, and a treatment-related mortality rate comparable to standard RIC regimens [111]. A phase I trial is currently ongoing at City of Hope to test the safety of a dose escalation of TMLI delivered before chemotherapy (Table 3).

*(c)* 
*Addition of TMI/TMLI in the conditioning regimen of haploidentical HSCT to reduce GvHD.*


TMI and TMLI may be added to the conditioning regimen of haploidentical HCT to enhance cytoreduction while mitigating GvHD without increasing the treatment-related mortality.

The preliminary results of a phase I study from City of Hope on 29 high-risk ALL, AML or MDS patients showed promising results with an induction regimen consisting of Flu (25 mg/m^2^/d, days −7 to −3), Cy (14.5 mg/Kg/d, days −7 and −6) and TMLI (ranging 12–20 Gy, days −7 to −3), followed by standard PTCy on days +3 and +4 (50 mg/kg). No increased toxicity was seen with a TMLI dose escalation to 20 Gy, while the OS and PFS rates were 83% and 76% at 1 year. The grade II-IV aGvHD rate was 61% and the treatment-related toxicity rate at 1 year was 9%. Based on these results, phase II studies are currently being planned [112].

*(d)* 
*Investigation of TMI/TMLI in standard-risk patients as an alternative to standard TBI.*


As described above, results from phase I studies have demonstrated the feasibility of integrating TMI/TMLI, eventually with a dose escalation, in combination with established chemotherapy regimens in high-risk patients and those with a poor prognosis. To date, no published reports are available for the investigation of the role of TMI/TMLI in standard-risk patients, but clinical trials have been launched in some institutions worldwide (Table 3).

*(e)* 
*TLI as nonmyeloablative conditioning*


With the aim of reducing toxicity and GVHD, while sparing efficacy, a nonmyeloablative conditioning regimen that uses the fractionated irradiation of the lymphoid tissues (TLI) and antithymocyte globulin (ATG) was introduced. The combination of TLI-ATG alters the host immune profile to favor regulatory natural killer T (NKT) cells that suppress GVHD by polarizing conventional T cells toward the secretion of noninflammatory cytokines such as IL-4 and by promoting the expansion of donor CD41 CD251 regulatory T cells.

In TLI, the radiation fields are directed to all major lymph-node-bearing areas, including the thymus and spleen, and are fractioned into daily low doses of 80–200 cGy each, resulting in minimal side effects.

The Stanford group initially reported the results of 37 patients with lymphoid malignancies or acute leukemias who underwent allo-HCT conditioned with TLI and ATG, describing a potent antitumor effect particularly in patients with lymphoid malignancies, with low GVHD complications and 36-month probabilities of OS and EFS of 60% and 40% [113,114].

The Gruppo Italiano Trapianto di Midollo Osseo (GITMO) conducted a prospective phase II clinical trial on 45 patients with hematological malignancies transplanted between 2007 and 2010, achieving similarly good results regarding disease control and the cumulative incidence of grade II to IV aGVHD and cGVHD [115].

The safety and tolerability of TLI-ATG emerged also in a study on 61 patients treated with allo-HCT for MDS, therapy-related myeloid neoplasms, MPN and chronic myelomonocytic leukemia [116].

A randomized phase II trial compared TLI-ATG to low-dose TBI (2 Gy) and fludarabine. This study showed that TLI-ATG was associated with a significantly lower risk of cGVHD (17.8% vs. 40.8%, *p*  =  0.017), but a higher risk of relapse (50% vs. 22%, *p*  =  0.017), leading to an equivalent OS at 4 years [117].

Recently, the Stanford group updated their single-center experience using TLI-ATG conditioning in a large cohort of patients (*n* = 612). The 1-year rate of NRM was 9%, while the incidences of aGVHD (grade II-IV) and extensive cGVHD were 14% and 22%, respectively. The 4-year OS and PFS were 42% and 32% for AML, 30% and 21% for MDS, 67% and 43% for CLL, 68% and 45% for NHL, and 78% and 49% for HL [118]. In conclusion, this latter study, as well as other single-center studies, suggest that TLI-ATG is a well-tolerated, non-myeloablative conditioning with a low risk of GVHD and NRM. Nevertheless, several studies are currently underway to reduce relapse rates while maintaining the favorable safety and tolerability profile of this regimen.

### 4.5. Ongoing Trials

Phase I–II trials are ongoing in several centers worldwide. To the best of our knowledge, no phase III trial is currently ongoing. Recently, clinical trials were launched also in patients in remission and with standard-risk disease, which could favor the replacement of current conditioning regimens with TMI/TMLI based regimens in selected patients. Table 3 summarizes the ongoing trials (listed at www.clinicaltrials.gov).

## 5. Conclusions and Future Directions

Previous studies have shown that TBI presents the following advantages over chemotherapy: (1) the eradication effect does not depend on the blood supply, with neither influenced by the inter-patient variability in drug absorption, metabolism, biodistribution, or clearance kinetics; (2) it can easily reach sanctuary sites such as the brain or testes; and (3) it provides a powerful means of immunosuppression in order to prevent the rejection of donor hematopoietic cells [119].

However, despite some clear pros, the delivery of conventional TBI is also associated with some concerns regarding toxicity, particularly in terms of its long-term cardiovascular and endocrine effects and potential to cause second tumors [50,72,82]. As a consequence, the use of TBI, at least from a classical perspective, is declining, with its use being questioned now also in ALL patients [27].

Lower TBI doses have been studied with the aim of reducing toxicity while maintaining efficacy, particularly in aged and less fit patients, with promising results; however, some concerns about disease control, especially in aggressive malignancies, remain.

In conclusion, a direct head-to-head comparison of TBI and the novel TMI/TLI approach has not been performed to date. However, the advantages of TBI are mainly now related to its long-term, standardized application in clinical practice, its efficacy in nearly all hematologic neoplasms, and well-known toxicity profile, particularly in pediatric patients.

On the other hand, the advantages of TMI/TLI techniques mainly rely on the following: (1) their conformational nature, allowing the toxicities to at-risk organs to be spared, thus improving the therapeutic index and tailoring of treatment; (2) the possibility of escalating the dose for target organs, particularly the bone marrow (especially in young, fit patients with advanced or high-risk disease); and (3) greater patient comfort and applicability to patients with physical limitations (since it is delivered in supine position). In time, TMI/TLI could potentially be demonstrated to improve both the safety and efficacy endpoints in HSCT recipients, as well as patient-reported outcomes; therefore, TMI/TLI might even replace the use of conventional TBI in several HSCT settings.

## Figures and Tables

**Figure 1 cancers-16-00865-f001:**
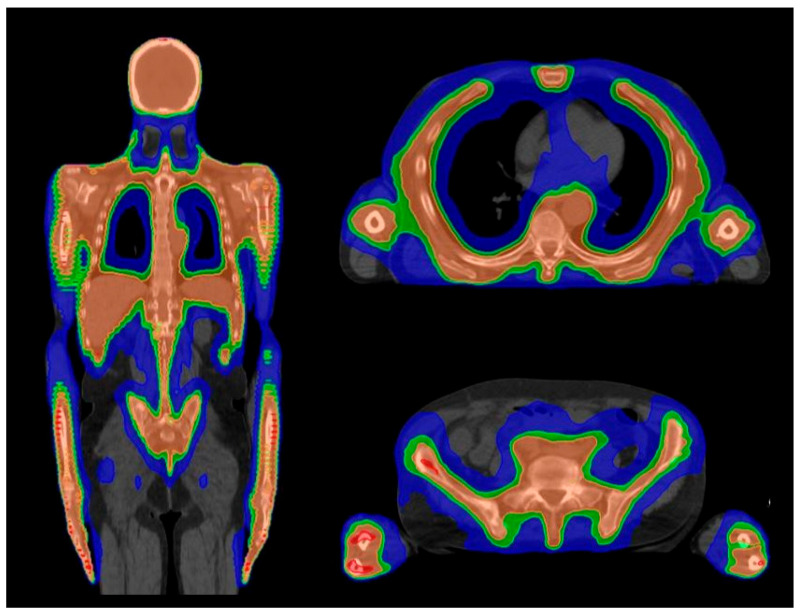
In blue 50% isodose, in green 80% isodose, in orange 95% isodose. (**Left**) Dose distribution of total body irradiation (TBI). (**Right**) dose distribution of total marrow and lymphoid irradiation (TMLI).

**Figure 2 cancers-16-00865-f002:**
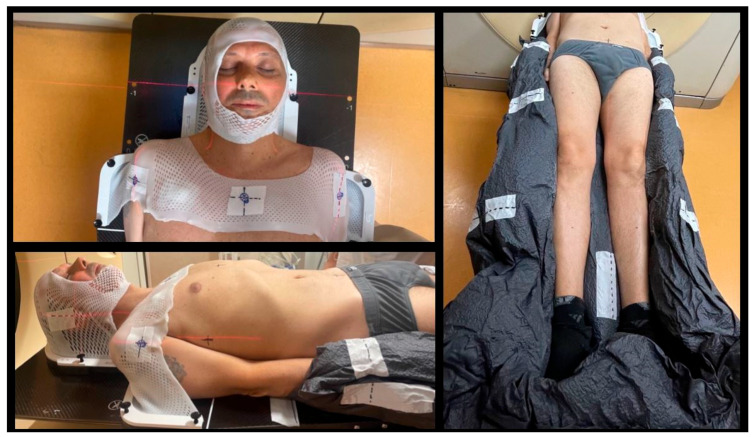
Immobilization devices for TMI/TMLI treatment. **Panel on the left**: thermoplastic mask for head and shoulders; **Panel on the right**: total-body vacuum pillow to immobilize thorax–abdomen and limbs.

**Figure 3 cancers-16-00865-f003:**
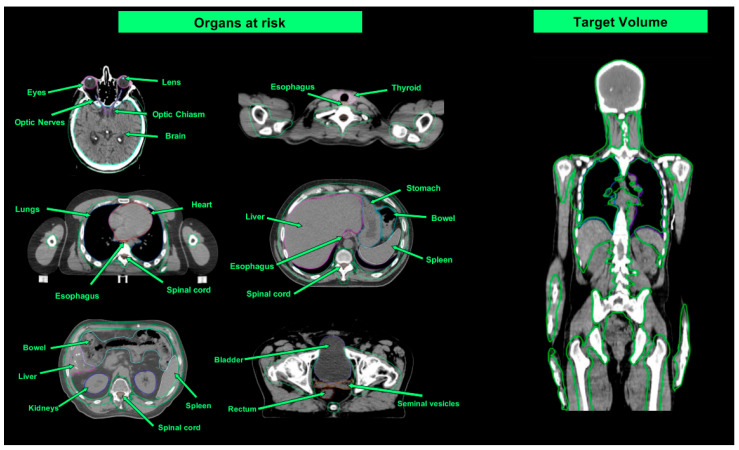
“Organs at risk”—representative axial slices of the organs at risk contoured in the planning phase of a TMI/TMLI treatment. “Target Volume”—representative coronal slice of the PTV contoured in the planning phase of a TMLI treatment.

**Table 2 cancers-16-00865-t002:** Median dose to at-risk organs with TBI compared to TMI/TMLI (representative studies and representative case from our institution).

Organ at Risk	TBI Median Doses (Gy)	Studies Evaluating TMI/TMLI Median Doses (Gy)		
	Wong et al. [51] (TBI 12 Gy)	Wong et al. [51] (TMI/TMLI 12 Gy)	Wong et al. [51] (TMI/TMLI 20 Gy)	Our Case (TMLI 12 Gy)
Brain	12.0	4.0	7.9	-
Lens	11.3	1.5	1.9	1.7
Eyes	11.3	6.6	7.0	5.7
Optic nerves	12.4	-	-	-
Oral cavity	11.8	3.9	4.8	8.5
Parotids	11.8	3.9	4.8	9
Thyroid	12.1	3.7	4.9	3.9
Esophagus	12.4	3.9	5.6	11.7
Breasts	11.5	6.9	8.7	-
Lungs	8.9	4.3	6.8	7.7
Heart	12.1	6.2	6.4	6.1
Stomach	12.2	3.1	5.0	5.5
Small Intestine	12.5	-	-	5.7
Liver	12.3	6.0	8.7	-
Kidneys	12.2	5.6	8.7	5
Bladder	12.4	7.0	7.4	6
Rectum	12.6	-	-	5.9

**Table 3 cancers-16-00865-t003:** Selected TMI and TMLI ongoing trials with acute leukemia, multiple myeloma or lymphoma patients (last check at www.clinicaltrials.gov in 1 January 2024).

Registration Number	Study Design	Type of HCT	Disease Type	RT Targets	TMI/TMLI Dose (Gy)	Chemotherapy Regimen	Estimated Enrollment
NCT02094794	Phase II	Allogeneic	AML and AML	Bone, spleen, nodes (full dose) Liver, brain (12 Gy)	20 (2 Gy fractions/BID)	Cy 100 mg/kg VP-16 60 mg/kg	87
NCT03467386	Phase I	Allogeneic	AML	Bone, spleen, nodes (full dose) Liver, brain (12 Gy)	18–20 (2 Gy fractions/BID)	*P-T* Cy 50 mg/m^2^/d × 2	24
NCT02446964	Phase I	Allogeneic Haploidentical	AML, ALL and MDS	Bone, spleen, nodes (full dose) Liver (12 Gy) Testes, brain (only ALL pts)	12–20 (1.5–2 Gy fractions/BID)	FLU 25 mg/m^2^/d × 5 Cy 14.5 mg/kg/d × 2 *P-T* Cy 50 mg/kg/d × 2	24
NCT03494569	Phase I	Allogeneic Haploidentical	Unfit or >55 years AML, ALL and MDS	Bone, spleen, nodes (full dose) Testes (only ALL pts)	12–20 (1.5–2 Gy fractions/BID)	FLU 30 mg/m^2^/d × 3 Mel 100 mg/m^2^ *P-T* Cy 50 mg/d × 2	36
NCT04262843	Phase II	Allogeneic Haploidentical	AML, ALL and MDS	TMLI	20 (2 Gy fractions/BID)	FLU (doses unknown) *P-T Cy* (doses unknown)	70
NCT03121014	Phase II	Allogeneic	High-risk AML and MDS	Bone	9 (1.5 Gy fractions/BID)	FLU 40 mg/m^2^/d × 4 BU 4800 uM/min	38
NCT02333162	Phase I	Allogeneic	Second HCT AML, ALL and MDS	Bone	N.A.	FLU + Mel	30
NCT03408210	NA	Allogeneic	AML, ALL and MDS	Total body or TMLI	TBI 10 Gy or TMLI 12–20 Gy	Cy 60 mg/kg/d × 2	191
NCT02122081	Pilot	Allogeneic	Unfit or >50 years	Bone	12 (2 Gy fractions/BID)	Cy (doses unknown)	45
NCT03262220	NA	Allogeneic	Unfit, age 40–80 Hematologic Malignancies	Bone	12 (4 Gy/die)	Variable schemes	87
NCT05139004	Phase I	Allogeneic	High-risk AML, ALL and MDS	Bone	TMLI 12 Gy	90Y-DOTA-anti-CD25 + FLU + Mel (doses unknown)	30
NCT00112827	Phase II	Autologous (tandem)	MM	Bone	16 (2 Gy fractions/BID)	Mel 200 mg/m^2^ for 1st auto-HCT	54
NCT02043847	Phase I	Autologous	MM relapsed/refractory	Bone	3–9 (3 Gy/fractions/die)	Mel 200 mg/m^2^	12
NCT00800059	Phase I/II	Autologous	MM	Bone	14–28 (2 Gy fractions/die)	N.A.	27

Abbreviations: Cy, cyclophosphamide; TMI, total marrow irradiation; TMLI, total marrow and lymphoid irradiation; Bu, busulfan; Flu, fludarabine; VP, etoposide, Treo, treosulfan; ALL, acute lymphoblastic leukemia; rALL, AML, acute myeloid leukemia; MDS, myelodysplastic syndromes; CML, chronic myeloid leukemia; MM, multiple myeloma. N.A., not applicable.

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
