# Peer review of "Maintain Efficacy and Spare Toxicity: Traditional and New Radiation-Based Conditioning Regimens in Hematopoietic Stem Cell Transplantation"

_cancers, 2024, doi:10.3390/cancers16050865_

Round 1

Reviewer 1 Report

Comments and Suggestions for Authors

The article offers a comprehensive exploration of radiation-based conditioning regimens in the context of hematopoietic stem cell transplantation (HSCT). The article presents a meticulous analysis of both traditional and innovative approaches to conditioning, aiming to balance efficacy with minimizing toxicity.

While the article discusses various radiation-based conditioning regimens, it could benefit from a more structured comparative analysis. Providing direct comparisons between traditional and modern approaches in terms of efficacy, toxicity profiles, and long-term outcomes would enhance the reader's understanding and facilitate informed decision-making in clinical practice.

Author Response

We thank the reviewer for his/her evaluation and general appreciation of our manuscript, and particularly for the suggestions made to improve it; as a proper, randomized comparative study of TMI/TLI vs standard TBI conditioning has not been performed so far, we have added to the conclusions section of the manuscript a paragraph comparing advantages and disadvantages of the two techniques to better clarify these issues.

We hope that the revised version of the paper might be now suitable for publication in Cancers.

Reviewer 2 Report

Comments and Suggestions for Authors

The review written by Dogliotti and colleagues addresses an important and currently much discussed topic in the field of hematopietic stem cell transplantation (HSCT): The question, how we could improve radiation-based myeloablative conditioning regimen.

Overall, the review is well written and I want to congratulate the authors on their solid work. It addresses the main aspects, which are:

1 leukemic disease control and how it might be maintained or improved with more advanced or modern techniques of radiotherapy as described in the manuscript. Conventional total-body irradiation (TBI) – based conditioning for HSCT has repeatedly demonstrated its high efficacy in relapse prevention despite its unfavourable late effects, which are of particular interest in pediatric patients but also in young adults. Therefore, it is highly appreciated if alternative, less toxic irradiation-based regimens could be developed. Pediatric-specific aspects were recently also reviewed by Hoeben et al (Hoeben B, Front Pediatr 2021).

I suggest to include also the FORUM trial (Ref 20 Peters C, JCO 2020) in Table 1 as it is the only prospective, randomized trials demonstrating the superiority of TBI/Etoposid vs chemoconditioning in pediatric ALL.

For a better reading I would suggest to shorten section 2 (TBI) of the manuscript.

The Reference 43 is in a wrong format, as it states the first names of the authors only. It should read: Spyridonidis A et al…

2 TBI- associated toxicity.

Most of the relevant points are discussed here, but the reader might lack somehow a specific statement on secondary malignancies including the increased brain tumor risk for children with ALL who have to undergo TBI-based allogeneic HSCT (Socie G, JCO 2000).

3 Novel developments in radiotherapy are described in detail in section 4 and 5 and provide a deep inside into alternative approaches including an overview on current trials.

Author Response

We thank the reviewer for his/her evaluation and positive comments about of our manuscript; we feel that the aspects underlined might help further improving the quality of the final paper.

In response to the comments made, we have included the FORUM trial in table 1 as suggested, corrected reference formats, shortened section 2 regarding TBI and added a specific paragraph regarding risk of secondary malignancies, including risk of brain tumor in ALL children. We hope that the revised version of the paper might be now suitable for publication in Cancers.